# Surface Topography in Cutting-Speed-Direction Ultrasonic-Assisted Turning

**DOI:** 10.3390/mi15060668

**Published:** 2024-05-21

**Authors:** Thanh-Trung Nguyen, Toan-Thang Vu, Thanh-Dong Nguyen

**Affiliations:** Precision Engineering and Smart Measurements Lab, School of Mechanical Engineering, Hanoi University of Science and Technology, Hanoi 100000, Vietnam; trung.nguyenthanh2@hust.edu.vn (T.-T.N.); thang.vutoan@hust.edu.vn (T.-T.V.)

**Keywords:** ultrasonic-assisted turning, cutting speed direction, intermittent cutting, continuous cutting, 3D surface topography, surface roughness

## Abstract

Ultrasonic vibration has been employed to assist in turning, introducing intermittent machining to reduce average cutting force, minimize tool wear, and enhance machining efficiency, thereby improving surface roughness. However, achieving intermittent cutting necessitates specific conditions, with a cutting speed or feed rate falling below the critical speed associated with the ultrasonic vibration parameters. This study presents a theoretical model for surface formation in cutting-speed-direction ultrasonic-assisted turning (CUAT), covering both continuous and intermittent machining regimes. Experimental validation was conducted on C45 carbon steel and 201 stainless steel to demonstrate the applicability of the theoretical model across different materials. Digital microscope analysis revealed 3D topography consistency with the theoretical formula. Surface roughness evaluations were performed for both CUAT and CT (conventional turning) methods. The results indicated a significant reduction in roughness Ra for C45 steel samples machined with CUAT, up to 80% compared to CT at a cutting speed of 20 m/min, while only exhibiting slight fluctuations when turning 201 stainless steel. Detailed analysis and explanation of these phenomena are presented herein.

## 1. Introduction

Ultrasonic-assisted machining is the process of applying ultrasonic vibrations to cutting tools or workpieces in traditional machining techniques such as turning [1,2], milling [3,4,5], drilling [6,7], grinding [8,9], and other methods [10]. The advantages of this method, including reduced cutting force and tool wear, as well as improved surface quality of machined parts, have been well documented in numerous empirical studies.

Turning is a prominent method in mechanical manufacturing; therefore, research aimed at improving product quality and reducing costs is crucial. Ultrasonic-assisted turning has received attention, with studies primarily focusing on various problems: designing and analyzing UAT structures using Finite Element Analysis (FEA) to optimize the technology system and energy use [11], experimentally evaluating improvements in roughness reduction and surface integrity [12,13], moderating tool wear while machining super-hard materials such as die steel and silicon carbide fiber-reinforced composites [14,15], and minimizing environmental impact by reducing the use of coolants and lubricants [16,17].

Ultrasonic vibration can be applied in one of three directions: cutting speed, feed, or radial. It can also be combined in two [18,19] and three directions [20,21]. Applying ultrasonic vibration in the cutting speed direction is more widely used [22,23] and effectively improves machining quality, especially when machining at cutting speeds below the critical value to achieve intermittent machining. Vibration in the radial [24] and feed [25] directions is often utilized to fabricate textured surfaces with specific properties, such as altered wettability, rather than focusing solely on improving machining efficiency. Both 2D and 3D vibration can significantly enhance surface quality; however, they require the meticulous design of horns and piezoelectric transducers and pose challenges in controlling vibration amplitude.

Research into surface formation during UAT is critical for understanding the enhancement of surface quality and the impact mechanism of ultrasonic vibrations. In one notable study [26], Schubert et al. investigated cutting force, surface roughness, and structure when turning with ultrasonic vibrations applied in all three directions: cutting speed, radial, and feed, with vibration amplitudes ranging from 1.5 to 4.9 µm. Although the experimental results presented are comprehensive, the explanation of surface integrity in CUAT remains incomplete, as the Rz roughness results for aluminum alloy AA2017 showed no improvement over CT at a cutting speed of 20 m/min. Further studies on surface formation in 2D and 3D UAT were also conducted by Guo et al. [27] and Lotfi et al. [28], respectively.

However, to unravel the mechanisms behind surface roughness fluctuation in CUAT, further experimental studies on surface formation are essential. This study aims to analyze the surface topography of workpieces machined using both CUAT and CT on C45 carbon steel and 201 stainless steel. The goal is to elucidate the impact mechanism of ultrasonic vibration on surface formation. Additionally, an analysis of surface roughness further reinforces the understanding of textured surface formation and highlights the advantages of CUAT.

## 2. Model of Intermittent Cutting in CUAT

In CUAT, the traditional cutting speed $v_{c}$ is augmented by the speed of ultrasonic vibration $v_{u}$, which is defined as 2πAfcos(2πft+φ). Hence, the aggregated cutting speed in CUAT is the sum of the traditional and ultrasonic cutting speeds, expressed as
(1)$$v_{a}=v_{c}+v_{u}=v_{c}+2\pi Afcos(2\pi ft+\varphi ).$$

It is evident that the ultrasonic vibration speed $v_{u}$ is not constant, and intermittent cutting occurs when [26,29]
(2)$$v_{c}\leq v_{crucial}=2\pi Af.$$

Conversely, if $v_{c}$ exceeds $v_{crucial}$, the aggregated cutting speed $v_{a}$ never reaches zero, resulting in continuous machining with the cutting speed varying between $v_{c}-2\pi Af$ and $v_{c}+2\pi Af$.

Figure 1 illustrates a schematic of ultrasonic vibration-assisted turning in the cutting speed direction (a), variation of aggregated cutting speed in intermittent cutting (b), and in continuous cutting (c). In the intermittent cutting process ($v_{c}\leq v_{crucial}$), ultrasonic vibration with frequency *f* assists the CT process having a traditional cutting speed $v_{c}$. During each vibration cycle, there is a period when no machining occurs, resulting in the formation of ridges on the machined surface. Along with the feed rate *fr*, the ridges are organized in an array similar to a grating pattern, with each ridge having a pitch Wi and a width equal to *fr*, as depicted in Figure 2. The cycle of this intermittent machining is characterized by
(3)$$T_{i}=\frac{10^{6}}{f}\;\left(\mu s\right)$$

Thus, the pitch of ridges on the workpiece surface is
(4)$$W_{i}=\frac{v_{c}}{60}\times \frac{10^{6}}{f}\;\left(\mu m\right)$$

For continuous cutting (where $v_{c}>2\pi Af$), changes to the surface also occur in a cycle similar to that of intermittent cutting; however, the key difference is that cutting speed never reaches zero. Instead, it fluctuates harmonically with an amplitude of *2πAf*. As a result, there is no distinct demarcation visible on the machined surface.

## 3. Experimental Setup

The ultrasonic vibration system used in the turning process consists of the ultrasonic generator KM-SS-2020 (K&M Technologies LTD, Shanghai, China), which has a power output of 2000 W, and the ultrasonic transducer F50-SA (Sino Sonics, Wuxi, China), operating at a frequency of 20 kHz with a peak-to-peak amplitude of 7 µm. The system also includes a horn that mounts the TCMT 16T308 (Dormer Pramet, Šumperk, Czech Republic) turning insert. This horn was specifically designed to match its natural frequency (including the turning insert and bolts) closely with the operational frequency of the transducer and generator. For the horn used in this experiment, the amplification factor was specified to be 2.8, resulting in a peak-to-peak vibration amplitude of approximately 19.6 µm.

Controlling vibration amplitude is important for evaluating the CUAT process, as it determines whether the machining behavior is intermittent or continuous. Therefore, the vibration amplitude was measured using a laser displacement sensor (Keyence, LK-H055, Osaka, Japan), set up as depicted in Figure 3a. The results were recorded in a CSV data file, capturing corresponding distances at designated sampling points. We then generated a distance graph based on these sampling points, as shown in Figure 3b. The sampling frequency used for the measurements was 392 kHz, approximately 20 times the frequency being measured. This setup made it straightforward to confirm the ultrasonic vibration period by simply multiplying the number of measurement points between two consecutive peaks or valleys by 2.55 µs.

CUAT experiments were conducted on a traditional lathe (OKK Ramo 1000, Yokohama, Japan), as depicted in Figure 4. The transducer was fixed on the tool post of the lathe at the horn’s neutral position. Workpieces made of C45 carbon steel and 201 stainless steel, with diameters ranging from 21 to 29 mm, were used for turning. The cutting speeds were set at 20, 40, 50, 60, 100, and 120 m/min, with a depth of cut at 0.15 mm and a feed rate of 0.1 mm/rev. Table 1 lists the experimental parameters for the CUAT process. The vibration amplitude and frequency were approximately 9.8 µm and 20 kHz, respectively ($v_{critical}=74\;m/min$). Under these conditions, CUAT exhibits both intermittent and continuous cutting behavior.

## 4. Results and Discussion

### 4.1. Surface Topography

To validate the theoretical model of machined surface formation in CUAT, surface topography was analyzed using a commercial digital microscope (Keyence, VHX 7000, Japan), equipped with a VH-ZST dual-objective zoom lens that offers magnification ranging from 20× to 2000×. Figure 5 and Figure 6 depict the surface topography of C45 steel and 201 stainless steel workpieces, respectively, machined at various cutting speeds below the critical speed. The regular ridges, corresponding to the intermittent machining periods are distinctly visible. Similarly to the textured surface formed by turning with ultrasonic vibration assisting in the feed rate [25] and radial [30] directions, the pitch of these ridges is calculated using Formula (4). However, the shape and arrangement of the ridges differ significantly because the workpieces were processed by Ultrasonic-Assisted Turning in the cutting speed direction.

When the cutting speed exceeds the critical speed, turning becomes continuous, and ridges no longer manifest on the machined surface. Instead, periodic crests are formed at the minimum aggregated cutting speed ($v_{c}-2\pi Af$). These crests, as in the intermittent cutting process, are uniformly distributed across the surface following a specific cycle. The pitch of the crests, calculated as $W_{i}=\frac{v_{c}}{60}\times \frac{10^{6}}{f}$, is consistently observed on both C45 steel and 201 stainless steel, as illustrated in Figure 7 and Figure 8.

By comparing the 2D and 3D images of the machined surface at cutting speeds above and below the critical speed, as illustrated in Figure 9, it can be observed that the ridges form during the machining stop, akin to the process of burr formation. Crests emerge at the minimum cutting speed, while at higher cutting speeds, the surface texture progressively transitions to a concave shape.

Upon comparing the theoretically calculated values of ridge pitch with measurements obtained from C45 and 201 stainless steel workpieces using a VHX-7000 series microscope, slight discrepancies were noted, as shown in Table 2 (where the average of pitches on the observed surface is represented as measured values). The largest standard deviations were 2.88 µm for 201 stainless steel and 3.7 µm for C45 steel, both at a cutting speed of 120 m/min, while at speeds below the critical speed, these standard deviations were smaller. This indicates that the influence of ultrasonic vibration remains relatively stable at low cutting speeds. However, when the cutting speed exceeds the critical value, the pitch errors of the ridges tend to increase.

### 4.2. Surface Roughness

Incorporating ultrasonic vibration into the machining process entails several intricate steps, including fabricating specialized horns and utilizing an ultrasonic generator. As a result, this integration reduces the flexibility of the CUAT process. Furthermore, the vibration of piezoelectric ceramics generates heat, and the utilization of electrical energy for vibration producing renders cooling during machining prohibitively expensive or impractical. Therefore, the adoption of CUAT must offer significant benefits to justify these challenges. Previous studies have demonstrated notable improvements in surface roughness. For instance, the authors in [31] showcased that surface roughness Ra was enhanced by up to 82% at a cutting speed of 60 m/min and a feed rate of 0.045 mm/rev when turning Al7Si0.3Mg alloy. Additionally, Bai [32] reported a reduction in roughness Ra from 1.8 to around 0.8 µm at a cutting speed of 20 m/min for Inconel 718.

The results of Ra roughness of C45 steel and 201 stainless steel workpieces at various cutting speeds in this experiment are presented in Figure 10. The most significant reduction in roughness, up to 80%, was observed on C45 steel at a cutting speed of 20 m/min. However, at speeds exceeding the critical speed, the improvement in roughness was less pronounced; at 100 and 120 m/min, Ra only decreased by 22% and 32%, respectively. For 201 stainless steel, Ra values showed minimal variation, with some measurements indicating higher roughness, while others were lower. Therefore, the observation of the 3D surface structure is necessary to provide objective assessments.

As shown in Figure 11a,c, the peaks in surface roughness of C45 steel in CT are typically caused by the traces of the turning insert, along with abnormal deformations due to fracture, poor chip evacuation, and chatter. The reduction in Ra roughness can be elucidated by examining the 3D images of the textured surface in CUAT, as illustrated in Figure 11b,d. At cutting speeds below the critical speed, ridges form after each vibration cycle, but their height remains relatively small. For example, at a cutting speed of 20 m/min, the highest peak in the 200 × 298 µm observation area was only 2.42 µm. Likewise, in the 400 × 586 µm region, at a cutting speed of 50 m/min, the peak height did not exceed 4.64 µm. Therefore, the textured surface with ridges and dimples created by ultrasonic vibration effectively mitigates the abnormal deformations and diminishes the height of the boundary between tool traces.

When the cutting speed surpasses the critical speed, ridge formation ceases as intermittent cutting is no longer achieved. Instead, continuous machining occured, with the aggregated cutting speed varying from $v_{c}-2\pi Af$ to $v_{c}+2\pi Af$. As a result, the surface exhibits waviness, with crests at the lowest cutting speed and troughs at the highest speeds. During the continuous machining process of CUAT, the difference in cutting speed, up to 4*πAf*, resulted in crests as high as 6.81 µm and 8.49 µm at cutting speeds of 100 m/min and 120 m/min, respectively, as depicted in Figure 12. The significant height of these crests, compared to the heights seen in CT, highlights a marked reduction in the influence of ultrasonic vibration at these higher speeds. Furthermore, at these high cutting speeds, abnormal deformations appear only minimally in CT, which enhances the surface roughness of the C45 steel workpieces. Consequently, the difference in roughness between CUAT and CT is relatively small.

The surfaces of 201 stainless steel workpieces also exhibit ridges when subjected to CUAT, as depicted in Figure 13a,c. However, during the CT process, there are almost no abnormal deformations, as seen in Figure 13b,d. Thus, the heights of the roughness peaks are primarily attributed to the cutting tool traces, indicating that the influence of ultrasonic vibration on 201 stainless steel material is negligible.

Consequently, the essence of the CUAT process lies in creating a textured surface that improves upon the one machined by CT. The formation of this textured surface critically influences surface roughness and is governed by the cutting speed (*v_c_*) and the vibration parameters such as amplitude (A) and frequency (f). During CUAT, intermittent cutting results in the formation of ridges and dimples: ridges are created as the tool transitions from a stop to cutting, akin to burr formation at the start of machining, while dimples result from the increased cutting speed induced by ultrasonic vibrations. To achieve a smooth surface with low roughness, the height difference between the ridges and dimples must be minimized. Calculating ridge height presents a challenge and heavily depends on the material properties. Conversely, the depth of the dimples correlates with the variations in aggregated cutting speed, as depicted in Figure 1b,c. To ensure minimal surface roughness, it is crucial to select appropriate values for *v_c_*, A, and f. However, these must be balanced with other factors such as material removal rate and chip evacuation conditions to optimize surface roughness.

## 5. Conclusions

CUAT experiments were conducted across cutting speeds ranging from 20 to 120 m/min, aiming to create textured surfaces characterized by ridges and crests. The surface topography of workpieces has been analyzed to delineate the formation of textured surfaces in both intermittent and continuous cutting modes. In intermittent cutting processes, ridges form periodically at intervals determined by the cutting speed and vibration frequency. For continuous cutting modes, crests form instead of ridges, following the same cycle observed in intermittent cutting. Typically, ridge heights are smaller than those of crests because ridges form during the transition from stopping to starting the cut, whereas crests develop primarily due to variation in cutting speed.

Throughout the tested cutting speed range, the surface roughness when turning C45 steel via CUAT was consistently lower than that in CT. The greatest decrease in roughness occurred at a cutting speed of 20 m/min, where it dropped from 2.88 to 0.58 µm, thanks to the effective removal of abnormal deformations by ultrasonic vibration. In contrast, the machined surface of stainless steel 201 in CT exhibited a notably smooth texture, basically influenced by the cutting tool traces without significant abnormal deformations. As a result, ultrasonic vibration exerted minimal impact, and the disparity in surface roughness (Ra) between CT and CUAT did not exceed 0.51 µm.

## Figures and Tables

**Figure 1 micromachines-15-00668-f001:**
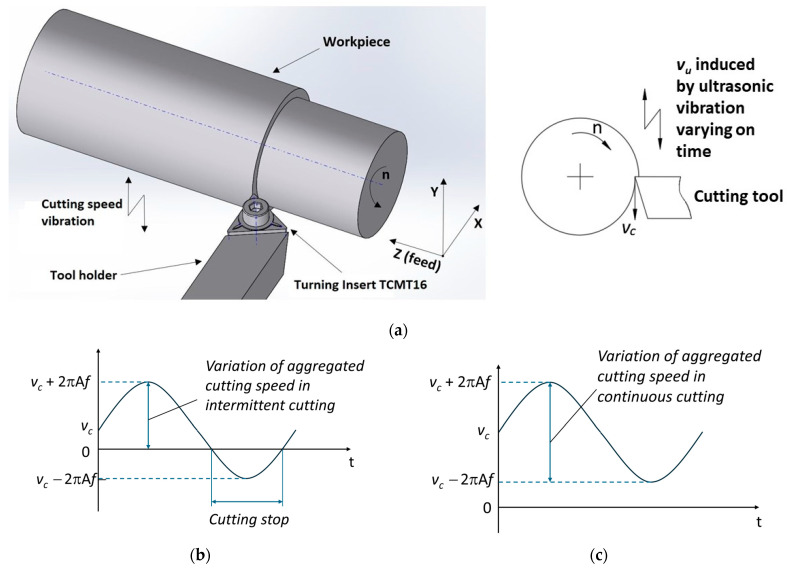
(**a**) Graphic outline of the CUAT process. (**b**) Variation of aggregated cutting speed in intermittent cutting. (**c**) Variation of aggregated cutting speed in continuous cutting.

**Figure 2 micromachines-15-00668-f002:**
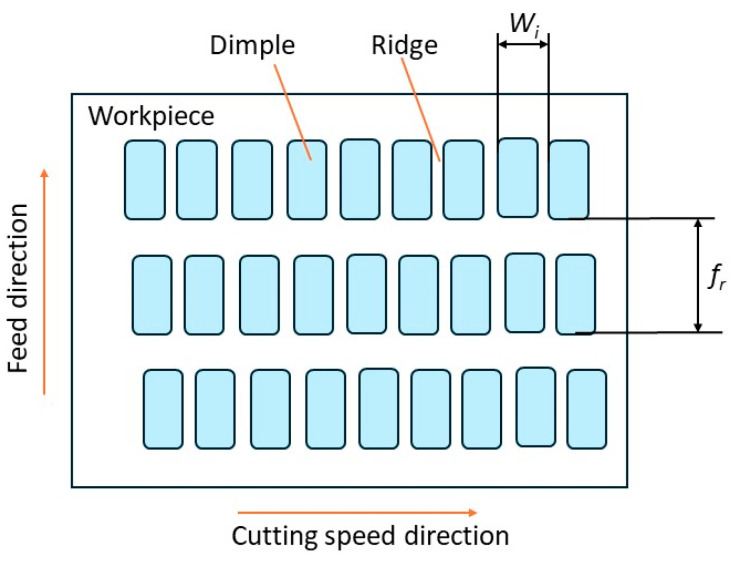
Model of periodical ridges on machined surface.

**Figure 3 micromachines-15-00668-f003:**
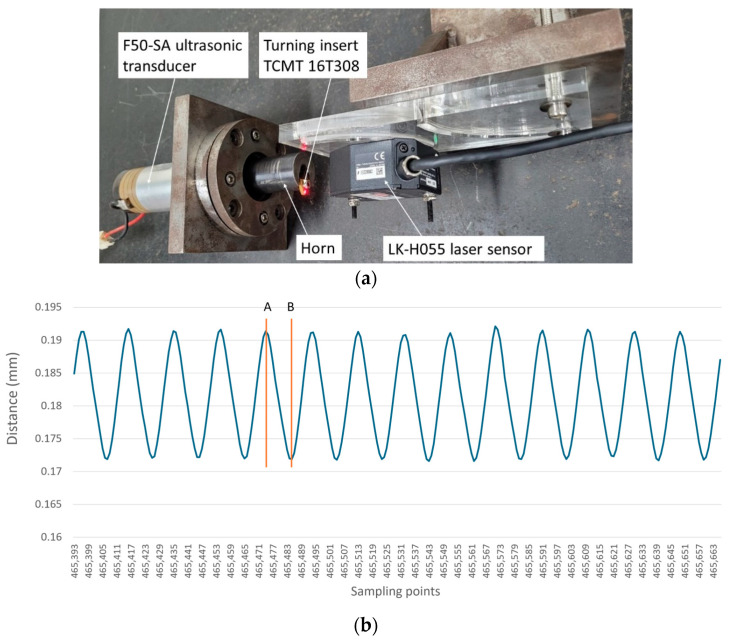
(**a**) Practical setup of vibration amplitude measurements. (**b**) The graph of distances with sampling points.

**Figure 4 micromachines-15-00668-f004:**
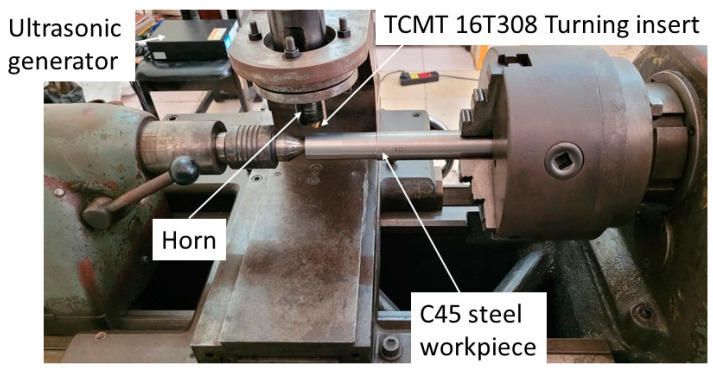
The CUAT process was conducted on a traditional lathe (OKK Ramo 1000, Japan).

**Figure 5 micromachines-15-00668-f005:**
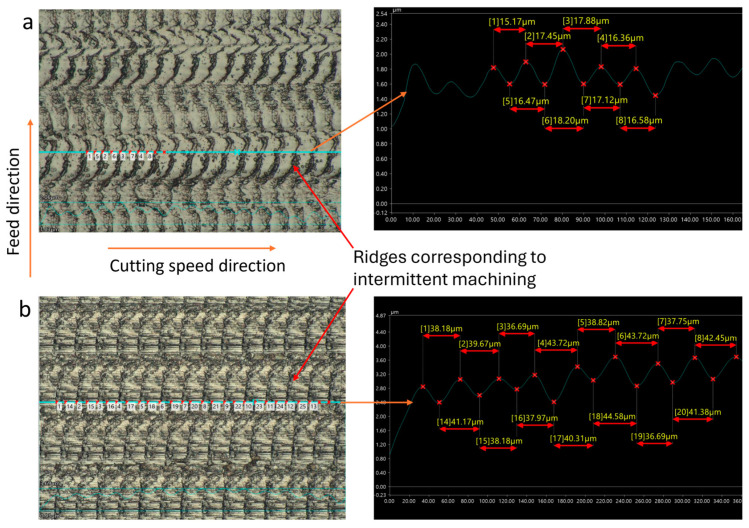
C45 steel surfaces fabricated by CUAT at cutting speeds of (**a**) 20 m/min, 1000× magnification, and (**b**) 50 m/min, 500× magnification.

**Figure 6 micromachines-15-00668-f006:**
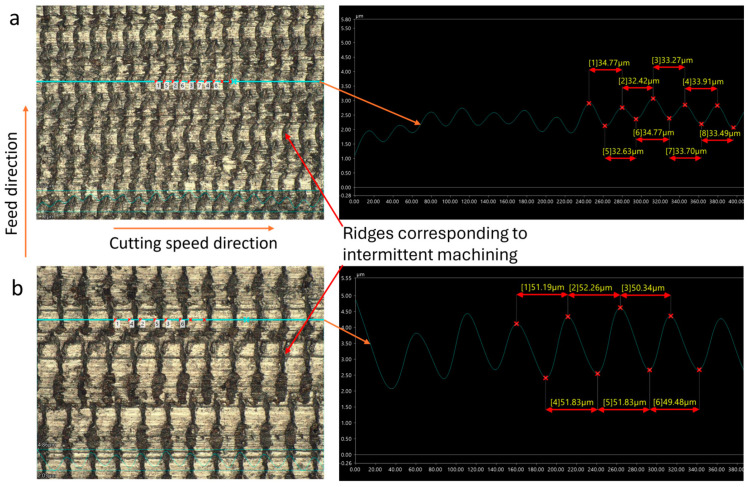
201 stainless steel surfaces fabricated by CUAT at cutting speeds of (**a**) 40 m/min, 500× magnification, and (**b**) 60 m/min, 500× magnification.

**Figure 7 micromachines-15-00668-f007:**
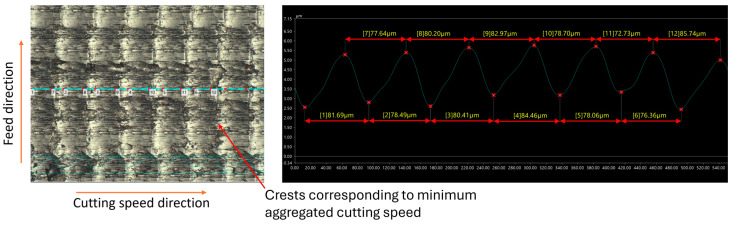
C45 steel surfaces fabricated by CUAT at a cutting speed of 100 m/min, 500× magnification.

**Figure 8 micromachines-15-00668-f008:**
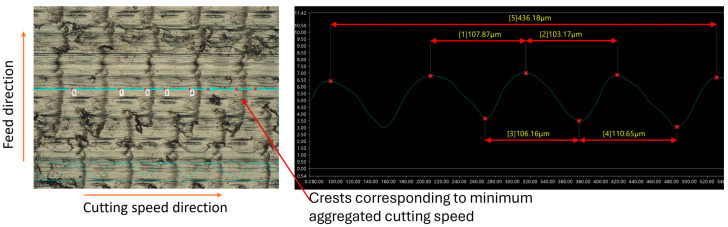
201 stainless steel surfaces fabricated by CUAT at a cutting speed of 120 m/min, magnification 500×.

**Figure 9 micromachines-15-00668-f009:**
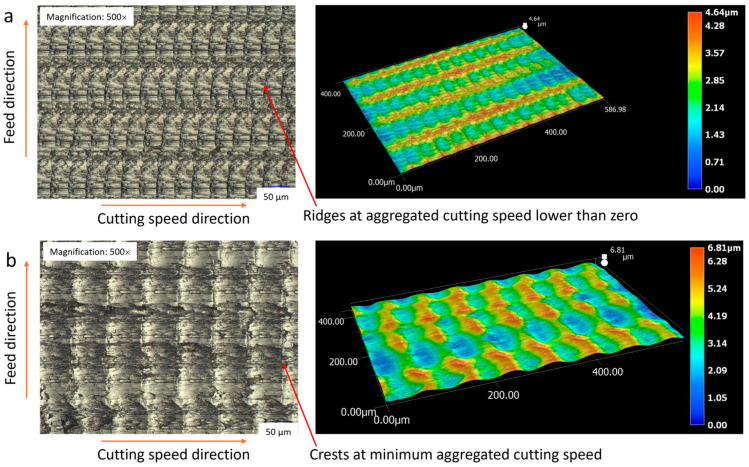
2D and 3D topography surfaces of C45 steel at cutting speeds of (**a**) 50 m/min, magnification 500×, and (**b**) 100 m/min, magnification 500×.

**Figure 10 micromachines-15-00668-f010:**
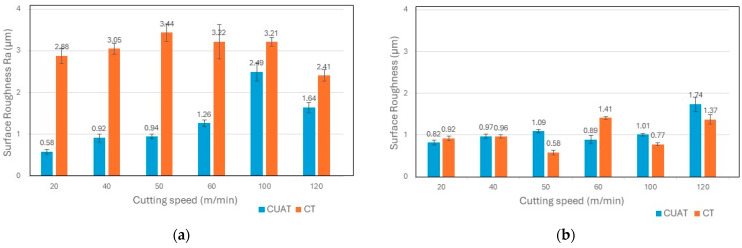
Surface roughness Ra at various cutting speeds for (**a**) C45 carbon steel and (**b**) 201 stainless steel.

**Figure 11 micromachines-15-00668-f011:**
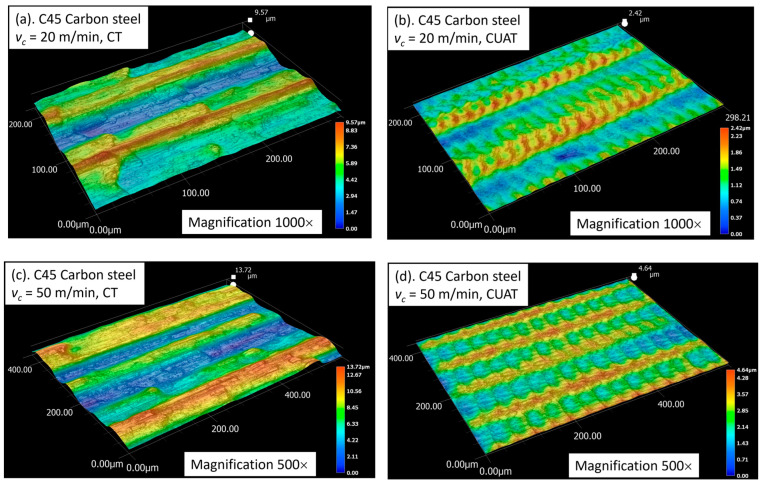
3D profile of C45 steel surface at (**a**) $v_{c}$ = 20 m/min in CT, (**b**) $v_{c}$ = 20 m/min in CUAT, (**c**) $v_{c}$ = 50 m/min, in CT, and (**d**) $v_{c}$ = 50 m/min in CUAT.

**Figure 12 micromachines-15-00668-f012:**
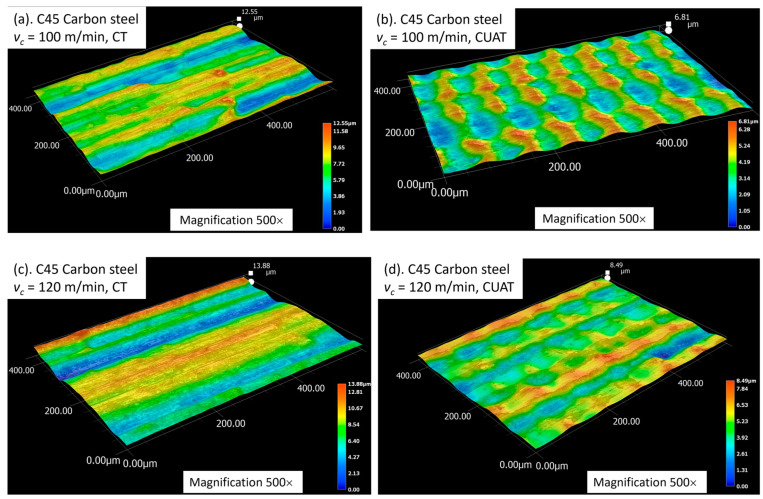
3D profile of C45 steel surface at (**a**) $v_{c}$ = 100 m/min in CT, (**b**) $v_{c}$ = 100 m/min in CUAT, (**c**) $v_{c}$ = 120 m/min in CT, and (**d**) $v_{c}$ = 120 m/min in CUAT.

**Figure 13 micromachines-15-00668-f013:**
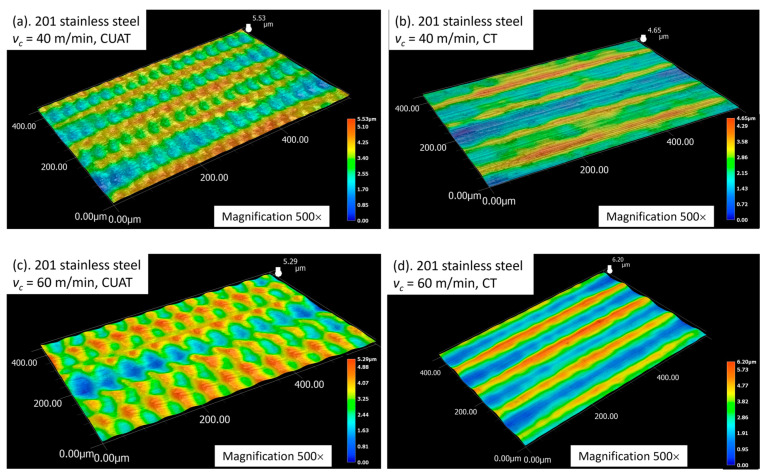
3D profile of 201 stainless steel surface at (**a**) $v_{c}$ = 40 m/min in CUAT, (**b**) $v_{c}$ = 40 m/min in CT, (**c**) $v_{c}$ = 60 m/min in CUAT, and (**d**) $v_{c}$ = 60 m/min in CT.

**Table 1 micromachines-15-00668-t001:** Experimental parameters for the CUAT process.

Workpiece	Material	C45 carbon steel; 201 stainless steel
	Diameter (mm)	21.8; 23.9; 27.6; 28.9; 29.3
Turning insert	Material	Tungsten carbide
	Rake angle	Positive
	Clearance angle (°)	7
	Nose radius (mm)	0.8
Vibration conditions	Amplitude (µm)	9.8
	Frequency (kHz)	20
Cutting conditions	Spindle speed (rev/min)	285; 550; 700; 1100; 1600
	Feed rate (mm/rev)	0.1
	Depth of cut (mm)	0.15

**Table 2 micromachines-15-00668-t002:** Pitch of ridges and crests on machined surface across theoretical calculation and practical measurement at 20 kHz ultrasonic vibration frequency.

Cutting Speed $v_{c}$ (m/min)	20	40	50	60	100	120
W_iTheory_ (µm)	16.67	33.33	41.67	50	83.33	100
W_iMeasurement_ for C45 steel (µm)	17.01	33.11	40.53	50.65	80.68	107.45
Deviation standard (µm)	1.24	1.00	1.58	1.49	2.53	3.70
W_iMeasurement_ for 201 stainless steel (µm)	16.77	34.09	39.33	51.49	77.14	107.39
Deviation standard (µm)	1.67	0.67	2.79	0.75	2.52	2.88

## Data Availability

Data is contained within the article.

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
