# Peer review of "Surface Topography in Cutting-Speed-Direction Ultrasonic-Assisted Turning"

_micromachines, 2024, doi:10.3390/mi15060668_

Round 1

Reviewer 1 Report

Comments and Suggestions for Authors

The article establishes a theoretical model of the surface formation mechanism of ultrasonic vibration-assisted turning in the direction of cutting speed from a geometrical point of view, and compares the surface morphology of C45 steel and 201 stainless steel under ultrasonic vibration cutting by comparing the surface morphology of C45 steel and 201 stainless steel, which is rich in the analysis of the experimental results, but the theoretical analysis of the article is not sufficiently informative, and the innovation of the research conducted is insufficient, the following are the modifications proposed for the content of the article, and the authors are requested to Please refer to the following.

1. In the abstract and section 4, the word ‘grating shape’ is not mentioned at all, but it appears in keyword, please select the appropriate words to revise keyword.

2. The introduction needs to be revised. Since the paper is an experimental paper, there are many references about device design in the introduction, which are not consistent with the content of the paper, and the references cited cannot reflect the purpose and significance of this paper.

3. At the end of the first section of the article, it is mentioned that "Currently, research on CUAT has not provided a complete explanation of the mechanism behind surface formation but attributes it primarily to intermittent machining", is it not rigorous enough to draw such a conclusion? To the best of our knowledge, there are already a number of studies on the formation mechanism of surface topography after ultrasonic vibration machining; moreover, the studies mentioned in this paper do not provide a systematic study on the surface formation mechanism, so please make an explanation or modification.

4. In the analysis of surface roughness, when the cutting speed is more than the critical cutting speed, compared with CT, surface roughness under CUAT is still reduced by 22% to 32%, but the paper does not give an explanation, please revise it.

5. The design of experimental parameters is missing from the Experimental setup, please add.

6. In section 3 of the article, it is mentioned that "The sampling frequency used in the measurements was 392 kHz, approximately 20 times the frequency being measured" Why not set the sampling frequency to a value close to the measured frequency? Please add relevant explanatory content.

Comments on the Quality of English Language

Moderate editing of English language required

Reviewer 2 Report

Comments and Suggestions for Authors

UVAT Surface morphology simulate has been studied by many researchers. Guo et al. and Liu et al. have established good method to simulate the UVAT Surface morphology. So, the expression "research on CUAT has not provided a complete explanation of the mechanism  behind  surface  formation" is improper.

In addition, since the UVAT Surface morphology can be simulated, the study of Part 2 " Model of intermittent cutting in CUAT " doesn't make much sense. Because the work in Part 2 only calculated the pitch of ridges on workpiece surface, which can be simulted by the work of Guo et al. and Liu et al. 

What's more, in the experiments, the pitch of ridges on workpiece surface calculated in Part 2 is not used to explain the surface roughness change mechanisms, proving that only calculating the pitch of ridges on workpiece surface is insignificant.

During the work for simulate the UVAT Surface morphology, the height of ridges must be calculated, not only the pitch!

Comments on the Quality of English Language

Acceptable

Reviewer 3 Report

Comments and Suggestions for Authors 1. The introduction is brief. More works on ultrasonic vibration machining should be covered, for example:
  • Surface roughness prediction in ultrasonic vibration-assisted milling (https://doi.org/10.1299/jamdsm.2020jamdsm0063)
  • Force prediction in ultrasonic vibration-assisted milling (https://doi.org/10.1080/10910344.2020.1815048)
2. Eq. (2) is not the only criterion for intermittent cutting. The speed only indicates that the gap between tool and workpiece is decreasing but not necessarily in contact. Please check the literature in comment #1 and comment. 3. Why is the depth of cut and feed rate fixed? 4. What is the standard deviation for measurements in Table 1? Same for Figure 10. 5. Is elastic recovery of material considered in theoretical model? That should improve the accuracy. 6. It doesn't seem that ultrasonic vibration shows significant benefits to improve the surface finish. Would a different vibration frequency or amplitude help? Please comment on it and do sensitivity analysis if needed.

Round 2

Reviewer 2 Report

Comments and Suggestions for Authors

none

Reviewer 3 Report

Comments and Suggestions for Authors

Comments are addressed properly.